# Pregnant women's attitudes towards complementary and alternative medicine and the use of phytotherapy during the COVID-19 pandemic: A cross-sectional study

**Aysegul Durmaz**[1]*, **Cigdem Gun Kakasci**[2]

**1** Department of Midwifery, Faculty of Health Sciences, Kutahya Health Sciences University, Kutahya, Türkiye, **2** Department of Midwifery, Faculty of Health Sciences, Suleyman Demirel University, Isparta, Türkiye

* aysegul.durmaz@ksbu.edu.tr

## Abstract

### Background

Approximately 80% of individuals worldwide use various holistic complementary and alternative medicine (HCAM) methods, including herbal products, to prevent diseases and improve their general health. In this study, it was aimed to investigate complementary and alternative therapy (CAM) and the use of phytotherapy by pregnant women in the COVID-19 pandemic period.

### Methods

This is a cross-sectional and descriptive study. The study included 381 women who applied to a public hospital in Türkiye and used herbal products during this pregnancy. Purposive sampling method was used. The study data were collected through "Identifying Information Form", "Holistic Complementary and Alternative Medicine Questionnaire (HCAMQ)" and "Information Form on the Use of Phytotherapy". In the analysis of the study data, descriptive statistics, the one-way ANOVA and multinomial logistic regression analyses were used.

### Results

The study was completed with 381 pregnant women. The average age, parity and gestational age of the pregnant women were 28.33±6.09, 2.17±0.95, 29.11±8.87, respectively. It was determined that 37.3% of pregnant women did not know the ingredients of the herbal products they used and 38.8% found them safer than the drugs. HCAMQ total mean score of the pregnant women was calculated as 34.62±16.22. It was found that the pregnant women used garlic the most (65.6%), followed by cumin (38.6%), curcuma (36.2%), and ginger (34.4%). HCAMQ total mean score was found to be lower in the pregnant women who found herbal products safer than drugs (p<0.001), who were not aware of the content of the herbal product they used (p<0.001), and who used herbal products so as to protect against COVID-19 (p = 0.041), to increase their physical resistance (p = 0.022), and to facilitate

**Data Availability Statement:** All relevant data are within the paper and its Supporting Information files.

**Funding:** The authors received no specific funding for this work.

**Competing interests:** The authors have declared that no competing interests exist.

childbirth (p = 0.002). It was determined that among the pregnant women who knew the content of the herbal products they used (Odds Ratio (OR) 1.122; 1.095–1.149 95%CI; p<0.001) and who did not know (OR 1.114; 1.085–1.144 95%CI; p<0.001), as negative attitude towards HCAM increased, their status of finding traditional drugs safer increased. Among the pregnant women who used herbal products to protect against COVID-19 (OR 1.142; 1.111–1.174 95%CI; p<0.001) and to increase their physical resistance (OR 1.120; 1.094–1.147 95%CI; p<0.001), as negative attitude towards HCAM increased, their status of finding conventional drugs safer increased.

## Conclusion

In today's world where the use of herbal products and CAM has become widespread, it is important to raise the awareness of pregnant women about the benefits and harms of these practices about which there is inadequate evidence.

## Introduction

Complementary and Alternative Medicine (CAM) is a broad term that includes interventions that are not part of the recognized standards in modern medical care [1, 2]. CAM methods are divided into five categories: mind–body therapies (e.g., meditation, hypnosis, imagery), biologically based practices (e.g., vitamins, dietary supplements, herbs), manipulative and body-based practices (e.g., massage therapy, reflexology, acupressure), energy healing (e.g., reiki, healing touch, therapeutic touch), and other complementary health approaches (e.g., Ayurveda, Chinese traditional medicine, acupuncture, naturopathic medicine) [2, 3].

Approximately 70% of low- and middle-income countries rely partially or entirely on the use of CAM to treat diseases [4]. Among WHO member states, 170 (88%) reported using CAM [5]. The use of CAM and herbal medicines are becoming increasingly common among pregnant women all over the world [6]. Nowadays, pregnant women turn to CAM and prefer herbal products to avoid the use of chemical products [7]. However, there is little scientific evidence regarding the effectiveness and safety of herbal product use during pregnancy. Additionally, there is clearly reported evidence about their negative effects in some pregnant women [8–11]. Nevertheless, pregnant women perceive herbal products as "natural" and therefore "safe" [12, 13]. Women's desire to use natural substances in pregnancy increases the popularity of herbal drugs. The rate of herbal product use in pregnancy was reported to range between 18% and 36% [7]. In a systematic review study, it was found that 47% of women used a herbal product at least once in their pregnancy [14]. CAM therapies are popularly (9–76%) applied in many countries [15]. It was reported that 69% of women in the US, 57% of women in the UK, and 51% of women in Germany used CAM methods [16]. In a study on the use of CAM methods in the pregnancy and breastfeeding periods, 92.8% of pregnant women were determined to use CAM methods. In the same study, the most frequently used therapies were determined to be herbal and massage therapies [15].

In many cultures, herbal products are used in pregnancy period in order to improve maternal and fetal health, alleviate gastrointestinal problems such as nausea, vomiting, and constipation, treat infections, to prepare for birth, and to initiate birth. The most frequently used herbal products in pregnancy period are ginger (*Zingiber officinale*), chamomile (*Matricaria chamomilla*), peppermint (*Mentha piperita*), echinacea (*Echinacea purpurea*), cranberries (*Vaccinium oxycoccus* and *Vaccinium macrocarpum*), garlic (*Allium sativum*), raspberry

(*Rubus idaeus*), valerian (*Valeriana officinalis*), fenugreek (*Trigonella foenum-graecum*), fennel (*Foeniculum vulgare*), and tea types (green and black tea) [17].

In a systematic review, almond oil (OR: 2.09; 95% CI: 1.07–4.08) use and heavy licorice (OR: 3.07; 95% CI: 1.17–8.05) use were associated with preterm labor, and the oral consumption of raspberry leaves (AOR: 3.47; 95% CI: 1.45–8.28) was associated with a higher likelihood of needing cesarean delivery. In the same study, the African herbal medicine mwanaphepo was associated with maternal morbidity (AOR: 1.28; 95% CI: 1.09–1.50) and neonatal death or morbidity [18]. It was revealed that commonly used herbal products such as ginger may have adverse perinatal outcomes [19]. Another study showed that pregnant women using traditional Chinese or Ayurvedic medicines may develop bone marrow depression and excessive bleeding [20]. Many countries, have no established guidelines for herbal products and their use. There is not enough scientific evidence for their use in pregnancy in the literature. Moreover, it is also important to identify the situations in which complementary-alternative medicine can be used [21, 22].

To the best of our knowledge, no studies have evaluated pregnant women's awareness of CAM and the information sources they use [2]. Additionally, there are a limited number of studies that specifically include pregnant women and question the purpose of using CAM [23]. Additionally, this study was conducted during the COVID-19 pandemic period. The COVID-19 pandemic brought restrictions on socializing and activities. It dominated daily life in Türkiye, as in many other countries around the world [24]. Studies on the impact of COVID-19 infection in pregnancy have shown that it increases the admission of pregnant women to intensive care units, the risk of mechanical ventilation, comorbidities such as preeclampsia and thrombosis, serious maternal and fetal complications, and deaths [25–28]. However, there have been inconsistencies between countries regarding vaccination. Additionally, recommendations about safety during pregnancy have constantly changed during the course of the pandemic. All these situations have caused panic, fear, and anxiety in pregnant women. Besides, many herbs and herbal products have been recommended without sufficient evidence of their safety. Misinformation has been disseminated frequently, especially on social media [29]. This situation has led pregnant women to prefer CAM methods to protect both themselves and their babies. Pregnant women usually consume herbal products without consulting health service providers. Many women use herbal products as they are natural without any concerns about the possibility of harming themselves or their fetus. Considering the widespread use of herbal products or CAM in pregnancy and lack of knowledge about how to use them, there is a need for more studies on their effects on human health. In this context, it is necessary to determine the status of pregnant women in terms of herbal product use status of pregnant women and their attitudes towards CAM.

In this study, it was aimed to investigate pregnant women's status of using complementary and alternative medicine in the COVID-19 pandemic process. The research questions to which answers were sought for were as follows:

1. What is the level of attitude among pregnant women regarding CAM?

2. What is the knowledge status of pregnant women about the herbal products they use?

3. What are the herbal products used by pregnant women?

4. For what purposes do pregnant women use herbal products?

5. Is there a relationship between pregnant women's reasons for using herbal products and their attitudes towards CAM and their status of finding herbal products safer than conventional drugs?

## Materials and methods

### Study design and setting

This cross-sectional, non-invasive, and descriptive survey study was conducted in Türkiye from March until May 2021 with women aged 18 and older who used herbal products during their current pregnancy. This study was carried out in a public hospital in Burdur in the Aegean Region of Türkiye. The hospital where the study was conducted provides healthcare to pregnant women 24 hours a day, 7 days a week. This study was presented according to the STROBE guidelines.

### Sample size and sampling method

According to the data of the Turkish Statistical Institute, there were 2491 live births in the province of Burdur in 2020 [30]. In the study, the purposive sampling method was used to reach pregnant women who used herbal products during their pregnancy. To reach the minimum required sample size calculated by power analysis, it was planned to include 734 pregnant women in the sample. The recruitment of participants continued until the targeted sample size was reached. As a result of power analysis, based on a Cohen's d effect size of 0.3 (Chi-squared test), with a margin of error of ($\alpha$) 0.01, and power of representing the population (confidence interval) at 99% (df = 5), the minimum sample size needed was calculated as 373. In order to reach the calculated sample number (with power analysis), 734 pregnant women were evaluated during the research. Recruitment of pregnant women continued until the sample size was completed. To compensate for potential data losses and increase the predictive power of the study and reduce its margin of error, the study was completed with 381 pregnant women. In the post-hoc power analysis performed, the decisions to be taken at the end of the study were found to be reliable and valid at 99.16% power.

### Participants

**Inclusion and exclusion criteria.** Pregnant women who volunteered to participate in the study between March until May 2021, lived in the Burdur province, used herbal products in their current pregnancy, were not diagnosed with a psychiatric disease and not receiving medications, were 18 years old or older, were literate in Turkish, and responded to the data collection forms completely were included in the study.

Pregnant women who did not use herbal products, refused to participate, or were admitted to the hospital due to an urgent health problem or birth were excluded from the study. Also, pregnant women who did not meet the inclusion criteria were excluded from the study.

**Data collection tool.** The study data were collected using an "Identifying Information Form", the "Holistic Complementary and Alternative Medicine Questionnaire (HCAMQ)", and an "Information Form on Phytotherapy Use".

*Identifying Information Form*: The form created by the researchers in line with the literature inquired about some sociodemographic characteristics (e.g., age, educational status, income level, employment status, family type, place of residence) and certain obstetric characteristics (e.g., parity, gestational week, planning status of pregnancy) [31–33].

*Holistic Complementary and Alternative Medicine Questionnaire (HCAMQ)*: The scale was developed by Hyland et al. (2003) [34], and its Turkish validity and reliability study was conducted by Erci (2007) [35]. The Likert-type scale has 11 items. The lowest and highest scores that can be obtained on the scale are 11 and 66. Higher scores indicate more positive attitudes towards CAM. In the study of Hyland et al. (2003), the Cronbach's Alpha coefficient of the

scale was calculated as 0.86 [34]. In the study conducted by Erci (2007) [35] and the our study, the Cronbach's alpha internal consistency coefficient of the scale was found to be 0.72.

*Information Form on Phytotherapy Use*: The information form included questions about the most frequently used herbal products (e.g., ginger, fennel, chamomile, peppermint-lemon, rosehip, echinacea, cumin, green tea) by the participants, the knowledge status of the participants regarding herbal products, their usage purposes, and their opinions [15, 17, 21].

**Data collection procedure.** The study data were collected by the researcher based on self-report. The data were obtained from pregnant women who presented to the relevant hospital for routine follow-ups (maternity outpatient clinic) or the Non-Stress Test using the face-to-face interview technique. Data collection process took approximately 10–15 minutes for each pregnant woman. To test the data collection tools, a pilot implementation was made with 20 pregnant women, and these pregnant women were not included in the sample. No herbal product use was recommended to the pregnant women during data collection period. Additionally, no guidance that might lead to a change of attitude towards complementary and alternative medicine was provided.

**Data analysis.** The collected data were analyzed by using the SPSS (22.0, IBM Corp., Armonk, NY) statistical package program. In the analyses of the data, descriptive statistics (mean, standard deviation, distribution range, frequency, percentage) were used. The data were found to have a normal distribution according to the results of the Kolmogorov-Smirnov test. In the analyses of the continuous variables, one-way ANOVA test and multinomial logistic regression analysis were performed. The threshold of statistical significance was determined as $p < 0.05$.

## Results

During the period of the study, 734 pregnant women were evaluated. 288 of the pregnant women were excluded from the study because they did not use herbal products, 8 were under the age of 18, 16 were illiterate in Turkish, and 23 refused to participate in the study. In addition, 18 pregnant women were excluded from the study because their data were missing. This study was completed with the participation of 381 pregnant women (Fig 1). The mean age of the participants was 28.33±6.09 years, the mean parity and gestational week was 2.17±0.95 and 29.11±8.87, respectively, and their mean total HCAMQ score was determined as 33.22±15.60. It was also determined that 40.4% of the pregnant women were high school graduates, that 77.7% had a nuclear family, that 65.6% were multiparous, that 66.1% were unemployed, and that 75.6% had planned their current pregnancy (Table 1).

It was found that 37.3% of the pregnant women were not aware of the contents of the herbal products they used, that 38.8% considered herbal products to be safer than drugs, that 22.6% had received information about herbal products from an herbalist, that 36.5% procured herbal products over the internet, and that 72.7% did not inform their midwife/physician about the herbal products they used. It was determined that 75.1% of the participants used herbal products to ensure protection against COVID-19, 82.7% used them to increase their physical resistance, 26.0% used them to relieve or alleviate hemorrhoid problems, 53.3% used them to treat constipation, 37.0% used them to fight insomnia, 57.7% used them to relaxation, 40.9% used them to facilitate childbirth, and 50.9% used them to positively affect their infant's health (Table 2).

It was seen that the majority of the pregnant women (65.6%) used garlic (*Allium sativum*), 38.6% used cumin (*Cuminum cyminum*), 36.2% used curcuma (*Curcuma longa*), and 34.4% used ginger (*Zingiber offinale*) (Fig 2).

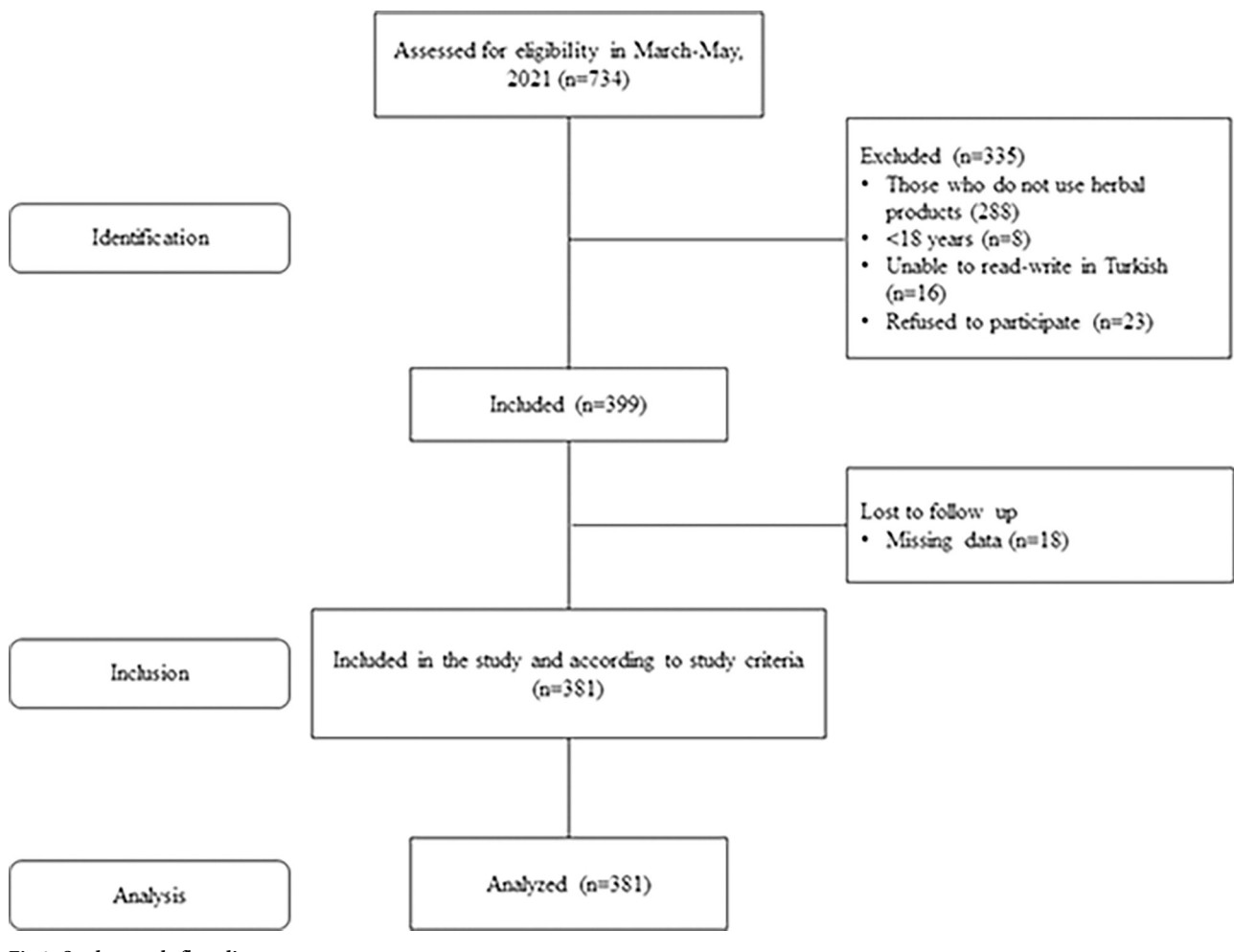

**Fig 1. Study sample flow diagram.**

The total mean HCAMQ scores of the pregnant women who considered herbal products to be safer than drugs (p<0.001), those who did not know the contents of the herbal products they used (p<0.000), those who used herbal products to protect themselves from COVID-19 (p = 0.041), those who used them to increase their physical resistance (p = 0.022) and those who used them to facilitate childbirth (p = 0.002) were found to be significantly lower. No statistically significant difference was determined between the pregnant women who used herbal products to positively affect their infant's health and those who used them for other reasons in terms of their mean total HCAMQ scores (p = 0.400) (Table 3).

According to the results of the logistic regression analysis, among the pregnant women who knew the contents of the herbal products they used, those with more negative attitudes towards HACM were 1.122 times (1.095–1.149 95%CI; p<0.001) more likely to find drugs safer than herbal products, and among the pregnant women who did not know about the contents of the herbal products they used, those with more negative attitudes towards HACM were 1.114 times (1.085–1.144 95%CI; p<0.001) more likely to find drugs safer than herbal products (Model 1). Additionally, it was found that among the pregnant women who used herbal products to protect themselves from COVID-19, those with more negative attitudes towards HCAM were 1.142 times (1.111–1.174 95%CI; p<0.001) more likely to find drugs safer than herbal products, while in pregnant women who used herbal products for reasons other than

**Table 1. Sociodemographic characteristics of the participants (n = 381).**

| Variables | | N | Mean±Sd | Distribution Range (Min-Max) | |
|---|---|---|---|---|---|
| **Age** | | 381 | 28.33±6.09 | 18 | 38 |
| **Parity (number)** | | 381 | 2.17±0.95 | 1 | 4 |
| **Gestational week** | | 381 | 29.11±8.87 | 8 | 42 |
| **Total mean HCAMQ score** | | 381 | 33.22±15.60 | 11 | 65 |
| Variables | N | % | Variables | N | % |
| **Educational level** | | | **Family type** | | |
| Primary-Secondary School | 139 | 36.5 | Nuclear | 296 | 77.7 |
| High School | 154 | 40.4 | Extended | 66 | 17.3 |
| Bachelor's degree or above | 88 | 23.1 | Broken | 19 | 5.0 |
| **Income status** | | | **Parity** | | |
| Income lower than expenses | 119 | 31.2 | Primigravida | 131 | 34.4 |
| Income equal to expenses | 218 | 57.2 | Multiparous | 250 | 65.6 |
| Income higher than expenses | 44 | 11.6 | | | |
| **Employment status** | | | **Pregnancy planning** | | |
| Employed | 129 | 33.9 | Planned | 288 | 75.6 |
| Unemployed | 252 | 66.1 | Unplanned | 93 | 24.4 |

Please check Table1 Data for variables descriptions.

protection from COVID-19, those with more negative attitudes towards HCAM were 1.103 times (1.077–1.129 95%CI; p<0.001) (Model 2). It was determined that in pregnant women who used herbal products in order to increase physical resistance, as negative attitude towards HCAM increased, their status of finding drugs safer than herbal products increased by 1.120 times (1.094–1.147 95%CI; p<0.001) more likely to find drugs safer than herbal products, while in the pregnant women who used herbal products for reasons other than increasing

**Table 2. Herbal product use characteristics of the participants (n = 381).**

| Variables | N | % | Variables | N | % |
|---|---|---|---|---|---|
| **Knows the contents of the herbal product she uses** | | | **Thinks herbal products are safer than drugs** | | |
| Knows the content | 239 | 62.7 | Herbal products are safer | 148 | 38.8 |
| Does not know the content | 142 | 37.3 | Drugs are safer | 233 | 61.2 |
| **Source of information about herbal products** | | | **Gets of herbal products from** | | |
| Internet | 61 | 16.0 | Pharmacy | 63 | 16.5 |
| Social media | 66 | 17.3 | Herbalist | 133 | 34.9 |
| TV | 55 | 14.4 | Internet | 139 | 36.5 |
| Family | 23 | 6.0 | Market | 46 | 12.1 |
| Friends | 38 | 10.0 | **Informs her midwife/physician about the herbal products she uses** | | |
| Health professionals | 52 | 13.6 | Informing | 104 | 27.3 |
| Herbalist | 86 | 22.6 | Not informing | 277 | 72.7 |
| **Reason for using herbal products during pregnancy[a]** | | | | | |
| Protection from COVID-19 | 286 | 75.1 | Increasing physical resistance | 315 | 82.7 |
| Relieve or alleviate of hemorrhoid problems | 99 | 26.0 | Treating constipation | 203 | 53.3 |
| Treatment of insomnia | 141 | 37.0 | Relaxation | 220 | 57.7 |
| To facilitate childbirth | 156 | 40.9 | Affecting the infant's health positively | 194 | 50.9 |

[a] Participants were allowed to mark multiple options.

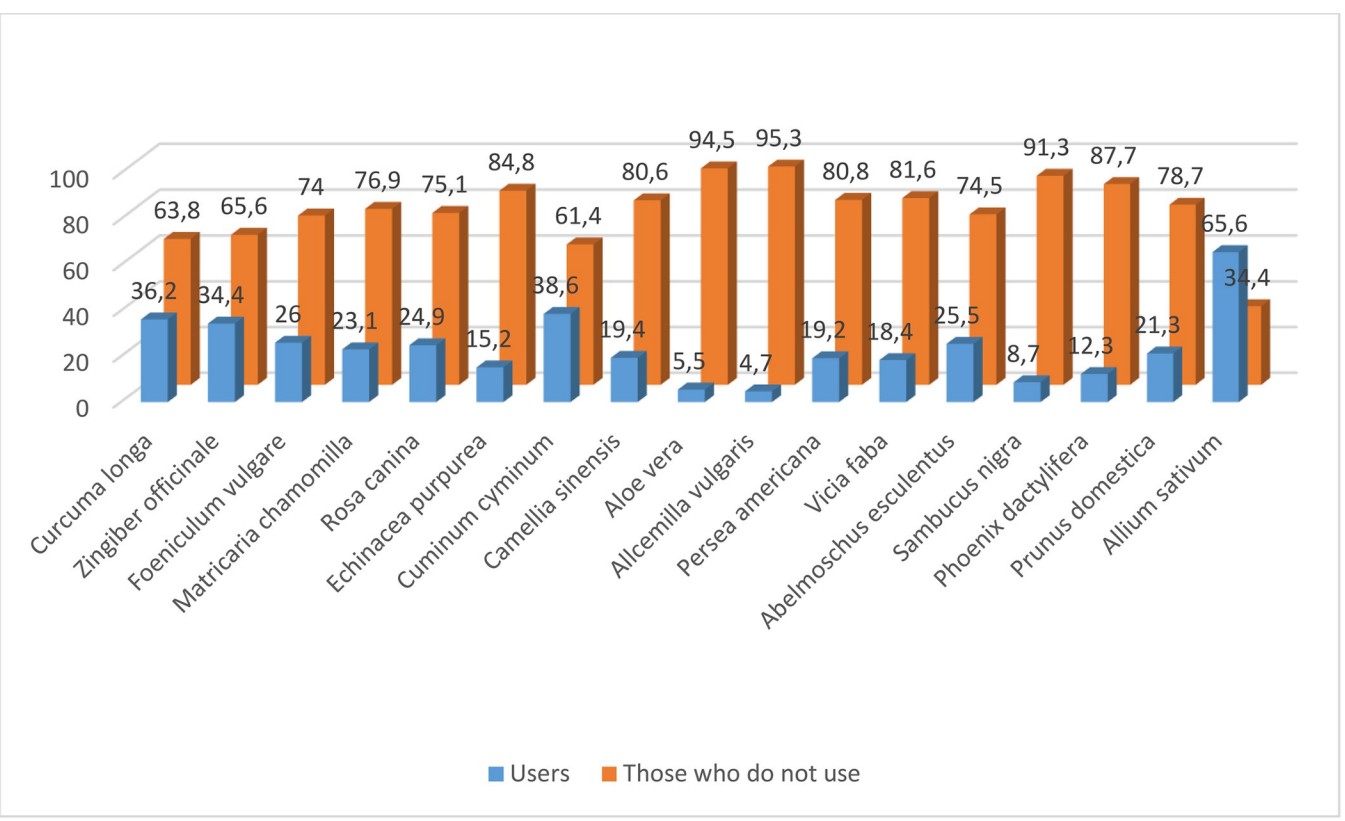

**Fig 2. Herbal products used by the pregnant women (n = 381) (Multiple options were allowed).** Please check S2 Table.

physical strength, those with more negative attitudes towards HCAM were 1.115 times (1.082–1.149 95%CI; p<0.001) more likely to find drugs safer than herbal products (Model 3) (Table 4).

## Discussion

In this study, the attitudes of pregnant women towards CAM and their use of herbal products were examined. In the study, it was found that the pregnant women had rather negative attitudes towards HCAM (33.22±15.60). Pregnant women who did not use herbal products, refused to participate, or were admitted to the hospital due to an urgent health problem or childbirth were excluded from the study. In a study in Türkiye, it was determined that pregnant women had negative attitudes towards HCAM (35.0±4.40) [36]. Another study also showed the negative attitudes of individuals towards HCAM (58.0±4.01) [37]. In yet another study, the majority of participants were found to have more attitudes than positive ones (above 33 points) towards HCAM [38]. On the other hand, in another study, no statistically significant difference was found in terms of HCAM-related attitudes between pregnant women who used herbal medicine and those who did not use herbal medicine [39]. In a web-based study, where the participants were from the UK, the US, Australia, Germany, Egypt, and China, while the Chinese participants were found to have positive attitudes towards HCAM (26.36 ±6.00), the Egyptian participants were determined to have negative attitudes (37.02±3.92) [40]. It was reported in e previous study that half of the pregnant women had a positive attitudes towards HCAM [41]. In another study conducted, the vast majority of the pregnant women (three-fourths) were determined to have a positive attitudes towards CAM [15]. Although

**Table 3. Comparison of the total mean HCAMQ scores and herbal products use characteristics of the pregnant women (n = 381).**

| Variables | HCAMQ | | |
|---|---|---|---|
| | | Statistics | |
| | Mean±Sd | t | p |
| **Thinking that herbal products are safer than drugs** | | | |
| Herbal products are safer than drugs | 22.07 ±10.70 | 14.338 | **0.000** |
| Drugs are safer than herbal products | 40.30 ±14.01 | | |
| **Knows the contents of the herbal product she uses** | | | |
| Knows the contents | 35.56 ±15.39 | 3.864 | **0.000** |
| Does not know the contents | 29.29 ±15.21 | | |
| **Protection from COVID-19** | | | |
| Uses herbal products to protect herself from COVID-19 | 32.24 ±15.23 | -2.065 | **0.041** |
| Uses herbal products for other reasons (to increase physical resistance, to relieve or alleviate hemorrhoid problems, to treat constipation, to treat insomnia, for relaxation, to facilitate childbirth, to positively affect the infant's health) | 36.18 ±16.38 | | |
| **Increasing physical resistance** | | | |
| Uses herbal products to increase her physical resistance | 32.25 ±14.84 | -2.335 | **0.022** |
| Uses herbal products for other reasons (to protect herself from COVID-19, to relieve or alleviate hemorrhoid problems, to treat constipation, to treat insomnia, for relaxation, to facilitate childbirth, to positively affect the infant's health) | 37.85 ±18.24 | | |
| **Facilitating childbirth** | | | |
| Uses herbal products to facilitate childbirth | 30.29 ±15.65 | -3.092 | **0.002** |
| Uses herbal products for other reasons (to protect herself from COVID-19, to increase physical resistance, to relieve or alleviate hemorrhoid problems, to treat constipation, to treat insomnia, for relaxation, to positively affect the infant's health) | 35.26 ±15.27 | | |
| **Positively affecting the infant's health** | | | |
| Uses herbal products to positively affect the infant's health | 32.56 ±15.59 | -0.842 | 0.400 |
| Uses herbal products for other reasons (to protect herself from COVID-19, to increase physical resistance, to relieve or alleviate hemorrhoid problems, to treat constipation, to treat insomnia, for relaxation, to facilitate childbirth) | 33.91 ±15.62 | | |

t; Independent-samples t-test

women are developing more positive attitudes towards HCAM around the world, the more negative attitudes of the pregnant women of this study in the pregnancy period, Which is thought to be risky in terms of the intake of drugs and herbal products can be considered as an expected result.

It was determined in the study that the majority of the pregnant women knew the contents of the herbal products they used, that they thought conventional drugs are safer than herbal products, they had received information about herbal products mostly from a herbalists, they bought herbal products mostly over the internet and most of them did not inform their midwife/physician about the herbal products they used. It was stated that the majority of the pregnant women (80.7%) had sufficient information about the herbal products they consumed [42]. It was reported that a large majority of the pregnant women used conventional drugs as

**Table 4. Regression analysis herbal products use and perceptions regarding herbal products use (N = 381).**

| | | B | Wald | Exp (B) | 95% confidence interval | | p |
|---|---|---|---|---|---|---|---|
| | | | | | Lower | Upper | |
| **Model 1[b]** | | | | | | | |
| **Thinks that herbal products are safer than drugs** | | | | | | | |
| Herbal products are safer than drugs[a] | | | | | | | |
| Drugs are safer than herbal products | Knows the contents of the herbal product she uses—HCAM | 0.115 | 86.607 | 1.122 | 1.095 | 1.149 | **0.000** |
| | Does not know the content of the herbal products she uses- HCAM | 0.108 | 63.677 | 1.114 | 1.085 | 1.144 | **0.000** |
| **Model 2[c]** | | | | | | | |
| **Thinks that herbal products are safer than drugs** | | | | | | | |
| Herbal products are safer than drugs[a] | | | | | | | |
| Drugs are safer than herbal products | Uses herbal products to protect herself against COVID-19 -HCAM | 0.133 | 88.879 | 1.142 | 1.111 | 1.174 | **0.000** |
| | Uses herbal products for reasons other than protection from COVID-19 -HCAM (to increase physical strength, to treat or alleviate hemorrhoid problems, to treat constipation, to treat insomnia, for relaxation, to facilitate childbirth, to positively affect the infant's health) | 0.098 | 67.074 | 1.103 | 1.077 | 1.129 | **0.000** |
| **Model 3[d]** | | | | | | | |
| **Thinks that herbal products are safer than drugs** | | | | | | | |
| Herbal products are safer than drugs[a] | | | | | | | |
| Drugs are safer than herbal products | Uses herbal products to increase physical resistance—HCAM | 0.114 | 87.671 | 1.120 | 1.094 | 1.147 | **0.000** |
| | Uses herbal products for reasons other than increase physical resistance -HCAM (to protect herself from COVID-19, to relieve or alleviate hemorrhoid problems, to treat constipation, to treat insomnia, for relaxation, to facilitate childbirth, to positively affect the infant's health) | 0.109 | 51.740 | 1.115 | 1.082 | 1.149 | **0.000** |

[a] Reference category

[b] Model 1 The effect of knowing the contents of herbal products used and attitude towards HCAM on finding drugs safer than herbal products

[c] Model 2 The effect of using herbal products to protect from COVID-19 and attitudes towards HCAM on finding drugs safer than herbal products

[d] Model 3 The effect of using herbal products to increase physical resistance and attitudes towards HCAM on finding drugs safer than herbal products

they considered them to be safer than herbal products [43]. Hwang et al. (2016) reported that while pregnant women were mostly influenced by friends, family, neighbors, and the media regarding their use of herbal products, their rates of being influenced by health professionals were lower [44]. It was obserced in another sample that pregnant women rarely informed their midwife or physician about the CAM they used [45]. In a systematic review of two studies, it was reported that the majority of the pregnant women in one study and approximately half of the pregnant women in the other study did not inform their physicians about the herbal products they used [6]. The results of these studies were consistent with the results of this study. What is interesting about the result of this study is that although the majority of the pregnant women did not find herbal products safe, all of them had used at least one herbal product. The COVID-19 pandemic, Which was an ongoing public health concern at the time when this study was carried out, may have encouraged the pregnant women to use herbal products.

The results of this study revealed that the pregnant women used herbal products mostly for protecttion from COVID-19, to increase physical resistance, to treat constipation, for relaxation, and to positively affect the infant's health. It was also found that the pregnant women used garlic, cumin, curcuma, and ginger the most. it was determined in a previous study that as pregnant women's knowledge levels of protective measures against COVID-19 decreased, their usage rates of herbal products increased [46]. According to the results of two systematic review studies, pregnant women used herbal products mostly for treating gastrointestinal disorders, common cold symptoms, and the flu symptoms. In the same studies, it was also

reported that the most frequently used herbal products were garlic, ginger, peppermint, thyme, fenugreek, chamomile, anise, sage, and green tea [6, 14]. In another study, the most frequently used herbal products by pregnant women were found to be thyme, peppermint, ginger, cumin, olive oil, cinnamon, chicory, and saffron. In the same study, the reasons of pregnant women for using herbal products were determined to be increasing physical resistance, improving fetal health, treating common cold/the flu symptoms or protection from the common cold/the flu, relaxation, treating constipation, and initiating labor [12]. In other study, some reasons for using herbal products during pregnancy were listed as constipation, sleep problems, anxiety, preparation for childbirth, and fatigue [7]. This study was conducted in a period when the COVID-19 pandemic was eat its peak. In this period, many myths and incorrect information spread among individuals (e.g., garlic preventing COVID-19). The pregnant women may have been affected by their potential exposure to inaccurate information in the aforementioned period and turned to herbal products.

The pregnant women who thought herbal products are safer than drugs, those who did not know the contents of the herbal product they used, those who used herbal products for protection from COVID-19, those who used them to increase physical resistance, and those who used them to facilitate childbirth had more positive attitudes towards HCAM. It was observed that the attitudes of the pregnant women towards HCAM were not significantly associated with whether they used herbal products to positively affect the health of their infants. Among both the pregnant women who knew the contents of the herbal products they used (1.122 times) and those who did not (1.114 times), those who had more negative attitudes towards HCAM were more likely to find conventional drugs safer than herbal products. Among pregnant women who used herbal products to ensure protection against COVID-19 (1.142 times) and among those who used them to increase physical resistance (1.120 times), those who had more negative attitudes towards HCAM were more likely to find conventional drugs safer than herbal products. In a previous study conducted, it was demonstrated that individuals had positive attitudes towards CAM in the context of protection from COVID-19 and increasing their body's immunity and resistance against the disease [37]. In another study, the pregnant women reported that they used CAM because they could maintain their personal control and saw it as a part of a holistic approach for health and well-being, and they displayed a positive attitude towards CAM. In the same study, the pregnant women stated that they could not directly get advice from their physicians and midwives or obtain adequate information from them [45]. In other study, it was determined that most of the pregnant women had a positive attitudes towards the use of herbal products for protection from COVID-19 [47]. In yet another study, the pregnant women who used herbal products considered CAM as a protective measure. However, in the same study, the women expressed their concerns about CAM regarding the availability of clinical evidence on the effectiveness of using these products [41]. In the relevant literature, it is seen that although women have a positive attitudes towards CAM, the lack of adequate evidence regarding the use of herbal products or CAM is a source of concern, and they needed to obtain more information about CAM from their midwife or physicians. This situation may explain the negative attitudes of women towards HCAM as their levels of knowledge about the use of herbal products in pregnancy increased.

## Strengths and limitations

In this study, which was conducted with the participation pregnant women who used herbal products, the herbal products that the pregnant women preferred to use in the COVID-19 pandemic period and general information about their reasons for using these products were investigated. One of the strengths of the study is that with the inclusion of only pregnant

women in the study, women's attitudes towards HCAM in a specific period as pregnancy were determined.

This study also had certain limitations. The data were collected based on self-reports of the participants, and illiterate pregnant women were excluded from the study. As only pregnant women who used herbal products were included in the study, the attitudes of pregnant women who did not consume herbal products could not be determined. In this study, the use of herbal products by the participants during their previous pregnancies was not evaluated. Hence, the attitudes towards HCAM that were identified in this study cannot be generalized to all pregnant women.

## Conclusion

In this present study, although the pregnant women were found to consume at least one herbal product, they were observed to display rather negative attitudes towards HCAM. It was seen that the participants who thought herbal products are safer than drugs, those who did not know the contents of the herbal products they used, those who used herbal products to protect themselves from COVID-19, those who used them to increase their physical strength, and those who used them to facilitate childbirth had more positive attitudes towards HCAM. More negative attitudes towards HCAM was observed in the pregnant women who knew the contents of the herbal products they used. Pregnant women should be definitely be provided with information about the effects of herbal product use and CAM in today's world where HCAM use is widespread and the probability of using HCAM in pregnancy irresponsibly has increased. Besides, education programs should be planned for pregnant women with positive attitudes towards HCAM. Additionaly, education on herbal products and CAM should be added to the content of prenatal care classes. Furthermore, in-service training should be provided to health professionals to encourage them to be more equipped in this regard. This way, health professionals can have information about evidence regarding the potential benefits and harms of herbal product use in pregnancy. Finally, as there is not enough evidence about herbal products, there is a need for randomized controlled trials to be conducted.

## Supporting information

**S1 Table.**
(XLSX)

**S2 Table.**
(DOCX)

## Author Contributions

**Conceptualization:** Aysegul Durmaz, Cigdem Gun Kakasci.

**Data curation:** Aysegul Durmaz.

**Formal analysis:** Aysegul Durmaz.

**Funding acquisition:** Aysegul Durmaz.

**Investigation:** Aysegul Durmaz, Cigdem Gun Kakasci.

**Methodology:** Aysegul Durmaz, Cigdem Gun Kakasci.

**Project administration:** Aysegul Durmaz.

**Resources:** Aysegul Durmaz.

**Software:** Aysegul Durmaz.

**Supervision:** Aysegul Durmaz, Cigdem Gun Kakasci.

**Validation:** Aysegul Durmaz, Cigdem Gun Kakasci.

**Visualization:** Aysegul Durmaz, Cigdem Gun Kakasci.

**Writing – original draft:** Aysegul Durmaz, Cigdem Gun Kakasci.

**Writing – review & editing:** Aysegul Durmaz, Cigdem Gun Kakasci.

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
