## [Decision Letter · Decision Letter 0]

25 Sep 2023

PONE-D-23-11358Investigation of complementary and alternative medicine and phytotherapy use in pregnancy in the covid-19 pandemic process: a cross-sectional studyPLOS ONE

Dear Dr. Durmaz

Thank you for submitting your manuscript to PLOS ONE. After careful consideration, we feel that it has merit but does not fully meet PLOS ONE’s publication criteria as it currently stands. Therefore, we invite you to submit a revised version of the manuscript that addresses the points raised during the review process. 1. Please consider revising the Title of your submission to better reflect the implications of your study. Since your study may apply to pregnancy in general, and not only during Covid?2. In your abstract kindly include the type of study, sample size, sampling methodology, dates and duration of the study and where the study was conducted? Also include pertinent results, implications, and relevant conclusions.3.  Kindly revise your introduction to show the aims and objectives of your study, and summarize the background in relation to other relevant studies in the literature.4. Kindly revise your materials and methods section to better describe the methods use including, sampling methodology, inclusion and exclusion criteria, and statistical methods, and assumptions. Also include information on ethical approvals, including information regarding consent by participants and methods used to maintain data and participants confidentiality..5. Please present all your findings in the results section.6. Kindly rewrite your discussion in the context of other studies in the literature, while focusing on the findings from your own research.7. Please revise your limitations section to better reflect the study limitations regarding generalization of the findings from this study.8. Please consider using and English language editor to address the grammatical and typographical errors in your manuscript before resubmission.9. Kindly address all other comments and observations as outlined by the peer-reviewers

We look forward to receiving your revised manuscript.

Kind regards,

Sylvester Chidi Chima, M.D., L.L.M, LLD.

Academic Editor

PLOS ONE

Journal Requirements:

2. Please upload a new copy of Figure 1 and 2 as the detail is not clear. Please follow the link for more information: " ext-link-type="uri" xlink:type="simple">https://blogs.plos.org/plos/2019/06/looking-good-tips-for-creating-your-plos-figures-graphics/"
https://blogs.plos.org/plos/2019/06/looking-good-tips-for-creating-your-plos-figures-graphics/"  

Reviewers' comments:

Reviewer's Responses to Questions

**Comments to the Author**

1. Is the manuscript technically sound, and do the data support the conclusions?

Reviewer #1: Partly

Reviewer #2: Yes

2. Has the statistical analysis been performed appropriately and rigorously? 

Reviewer #1: Yes

Reviewer #2: Yes

3. Have the authors made all data underlying the findings in their manuscript fully available?

Reviewer #1: Yes

Reviewer #2: Yes

4. Is the manuscript presented in an intelligible fashion and written in standard English?

Reviewer #1: No

Reviewer #2: Yes

5. Review Comments to the Author

Reviewer #1: This manuscript could be an important reference for future studies. However, is still needed to improve the quality of this paper. Please revise the manuscript to address the expressed concerns. After thorough review, I am recommending some revisions. In this regard, kindly address the following comments and suggestions to further improve your manuscript

a. There are some spelling and grammatical errors in the text. Please correct them

b. The quality of the images used is very low

c. Please write the type of study, sample size, sampling strategy and date and country of study in abstract

d. It was better if you wrote some of main finding as quantitative or mean ±SD within the abstract. The result section in the abstract is poor and immature!!

e. The Introduction is so weak. You could summarize this section a bit more for readers. Write about the problems, the novelty of your study, and your study goals within the introduction. The limitations of prior research might also be mentioned by the authors as further support for their present investigation. In this section, you can use the following articles:

1- “ Perceptions and personal use of Complementary and Alternative Medicine (CAM) by Iranian health care providers ”

2- “Use and perception of complementary and alternative medicine among Iranian psychologists”

3- “The predictors of the use of complementary and alternative medicine among type 2 diabetes patients based on the health belief model”

f. The materials methods section is relatively immature. You could expand it a bit more clearly for readers. For example, where have you collected samples? Write the year and the name of place in which you had done this survey. Furthermore, write about all applied exclusion and inclusion criteria a bit more clearly by which you selected samples for this survey.

g. The methods need to be improved by providing more detail information related to participant’s selection (e.g. respond rate; necessary permissions from who? How did the researcher contact the potential participants?)

h. Describe the validity and reliability of the measurement tool

i. Mention the possible score (range) for each scales and meaning of it so easier to readers interpret the results.

j. What are the data extract’s center characteristics? is it governmental or private, is it referral or not referral and so on, discuss more about it

k. Discuss more about your sampling strategy? The structure of your sampling is so vague and understandable. Did you have sampling frame? how did you access to this frame

l. In the discussion section I would like to see a more profound discussion about the findings. What is the meaning of your results in light of earlier studies. The discussion should be more than only a repetition of the results accompanied by some arbitrary studies. in sum, the authors need to state how the research fills an identified knowledge gap and how the new knowledge will be used to improve population

Reviewer #2: - In the introduction, it is stated that during pregnancy, women use herbal products for various reasons and accept them more than chemical products, so why is it necessary to do your research? The reason for conducting this research is not justified at all. Why during the Covid era? What effect does it have on pregnancy? Why are herbal products more favored by pregnant women? Similar studies? The introduction is poorly written. It needs a general revision.

- There is no mention of the method of sampling in the working method. Why was this number of samples selected? On what basis? - In the results section, there is no discussion about the main purpose of the research. Why were these drugs used during the corona epidemic? The rest of the results obtained in other studies. The effect of these plants against Corona and the importance of their use should be discussed first. Then see if it is used or not.

- Regarding the attitude of pregnant women, why has there been a change in attitude towards herbal products or medicines? You, who have not provided training, have investigated this change in attitude. How come the attitude has changed and the results have been mentioned in this way, please explain clearly.

- Please write the discussion part coherently. Write the results of your study, the positive and negative studies related to that conclusion in one place and discuss it.

- In the limitations section, you have stated that the plants that they preferred to use during the corona period were examined, but this was not stated during the study, and only apparently the study was conducted during this period, but whether people did not use it before and now only In this era, these items are used only to protect against the corona virus. It is not mentioned. It is better for multiparous women who did not use these items in their previous pregnancies and now use these items because of the corona virus. were discussed more.

6. PLOS authors have the option to publish the peer review history of their article (what does this mean?). If published, this will include your full peer review and any attached files.

Reviewer #1: **Yes: **Dr. Hadi Tehrani

Reviewer #2: **Yes: **Parvin Mangolian shahrbabaki

---

## [Author Response · Author response to Decision Letter 0]

9 Nov 2023

Reviewer 1 

Suggestion: This manuscript could be an important reference for future studies. However, is still needed to improve the quality of this paper. Please revise the manuscript to address the expressed concerns. After thorough review, I am recommending some revisions. In this regard, kindly address the following comments and suggestions to further improve your manuscript.

Thank you for taking your valuable time to evaluate our article and for your contribution to the article. Corrections made in line with your suggestions are indicated in the main text in red.

Suggestion a: There are some spelling and grammatical errors in the text. Please correct them

Correction a: Thank you for your suggestion. The text was checked for spelling and grammatical errors. Corrections have been made.

Suggestion b: The quality of the images used is very low

Correction: We used 300 dpi resolution as in the figure preparation rules. However, the quality of the image is still quite low. The quality of the image has been increased.

Suggestion c: Please write the type of study, sample size, sampling strategy and date and country of study in abstract

Correction c: Thank you for your suggestion. Recommended information has been added in the abstract section.

Suggestion d: It was better if you wrote some of main finding as quantitative or mean±SD within the abstract. The result section in the abstract is poor and immature!!

Correction d: Thank you for your suggestion. We expanded the results section in the abstract.

Suggestion e: The Introduction is so weak. You could summarize this section a bit more for readers. Write about the problems, the novelty of your study, and your study goals within the introduction. The limitations of prior research might also be mentioned by the authors as further support for their present investigation. In this section, you can use the following articles:

1- “ Perceptions and personal use of Complementary and Alternative Medicine (CAM) by Iranian health care providers ”

2- “Use and perception of complementary and alternative medicine among Iranian psychologists”

3- “The predictors of the use of complementary and alternative medicine among type 2 diabetes patients based on the health belief model”

Correction d: Thank you for your suggestion. The introduction section has been expanded according to your suggestions.

Suggestion f: The materials methods section is relatively immature. You could expand it a bit more clearly for readers. For example, where have you collected samples? Write the year and the name of place in which you had done this survey. Furthermore, write about all applied exclusion and inclusion criteria a bit more clearly by which you selected samples for this survey.

Correction f: Thank you for your suggestion. In the method section, explanations of the study design, setting, sample size and sampling method have been expanded.

Suggestion g: The methods need to be improved by providing more detail information related to participant’s selection (e.g. respond rate; necessary permissions from who? How did the researcher contact the potential participants?)

Correction g: Thank you for your suggestion. In the method section, explanations of the sampling method and participants have been expanded. Information about permissions is explained in the “Ethical aspect of the study” section.

Suggestion h: Describe the validity and reliability of the measurement tool

Correction h: Thank you for your suggestion. Added suggested information.

Suggestion i: Mention the possible score (range) for each scales and meaning of it so easier to readers interpret the results.

Correction i: Thank you for your suggestion. The requested information is included in the text. "The lowest and highest scores to be obtained from the scale are 11 and 66. As the score on the scale decreases, positive attitude towards complementary and alternative medicine increases."

Suggestion j: What are the data extract’s center characteristics? is it governmental or private, is it referral or not referral and so on, discuss more about it

Correction j: Thank you for your suggestion. Data were collected from a public hospital. There was no guidance from anyone or the system to collect the data. Among the pregnant women who admitted to the hospital, those who met the inclusion criteria were included in the study. Since this information was mentioned in the method section, it was not added to the procedure section because it would be repetitive and disrupt the flow.

Suggestion k: Discuss more about your sampling strategy? The structure of your sampling is so vague and understandable. Did you have sampling frame? how did you access to this frame

Correction k: Thank you for your suggestion. In the method section, explanations of the study design, setting, sample size and sampling method have been expanded. Since the article is written according to STROBE, detailed information is given at the beginning of the results section.

Suggestion l: In the discussion section I would like to see a more profound discussion about the findings. What is the meaning of your results in light of earlier studies. The discussion should be more than only a repetition of the results accompanied by some arbitrary studies. in sum, the authors need to state how the research fills an identified knowledge gap and how the new knowledge will be used to improve population

Correction: Thank you for your suggestion. The discussion section has been rearranged in line with your suggestions.

Reviewer 2

Thank you for taking your valuable time to evaluate our article and for your contribution to the article. Corrections made in line with your suggestions are indicated in the main text in blue.

Suggestion: In the introduction, it is stated that during pregnancy, women use herbal products for various reasons and accept them more than chemical products, so why is it necessary to do your research? The reason for conducting this research is not justified at all. Why during the Covid era? What effect does it have on pregnancy? Why are herbal products more favored by pregnant women? Similar studies? The introduction is poorly written. It needs a general revision.

Correction: Thank you for your suggestion. A general revision was made in the introduction section.

Suggestion: There is no mention of the method of sampling in the working method. Why was this number of samples selected? On what basis? 

Correction: Thank you for your suggestion. In the method section, explanations of the study design, setting, sample size and sampling method have been expanded.

Suggestion: In the results section, there is no discussion about the main purpose of the research. Why were these drugs used during the corona epidemic? The rest of the results obtained in other studies. The effect of these plants against Corona and the importance of their use should be discussed first. Then see if it is used or not.

Correction: Thank you for your suggestion. In the study, it was aimed to investigate the CAM and the use of herbal products by pregnant women in the COVID-19 pandemic process. For this reason, in the study, pregnant women who used herbal products were questioned about which plants they consumed and for what purpose. The effect of herbal products on Covid-19 has not been evaluated. The discussion on herbal products has been expanded.

Suggestion: Regarding the attitude of pregnant women, why has there been a change in attitude towards herbal products or medicines? You, who have not provided training, have investigated this change in attitude. How come the attitude has changed and the results have been mentioned in this way, please explain clearly.

Correction: Thank you for your suggestion. In this study, the current CAM attitude levels of pregnant women were examined. No training or initiative has been taken to cause pregnant women to change their attitudes. Changes in attitudes (pre test - post test) were not evaluated.

Suggestion: Please write the discussion part coherently. Write the results of your study, the positive and negative studies related to that conclusion in one place and discuss it.

Correction: Thank you for your suggestion. The discussion section has been rearranged in line with your suggestions.

Suggestion: In the limitations section, you have stated that the plants that they preferred to use during the corona period were examined, but this was not stated during the study, and only apparently the study was conducted during this period, but whether people did not use it before and now only In this era, these items are used only to protect against the corona virus. It is not mentioned. It is better for multiparous women who did not use these items in their previous pregnancies and now use these items because of the corona virus. were discussed more.

Information that the study was conducted during the COVID-19 period has been added to the article. The following sentence has been added to the limitations section. 

“Pregnant women's use of herbal products in their previous pregnancies was not evaluated.”

---

## [Decision Letter · Decision Letter 1]

13 Dec 2023

Pregnant women's attitudes towards complementary and alternative medicine and the use of phytotherapyduring the COVID-19 pandemic: A cross-sectional study

PONE-D-23-11358R1

Dear Dr. Durmaz,

We’re pleased to inform you that your manuscript has been judged scientifically suitable for publication and will be formally accepted for publication once it meets all outstanding technical requirements.

Kind regards,

Sylvester Chidi Chima, M.D., L.L.M.

Academic Editor

PLOS ONE

Reviewers' comments:

Reviewer's Responses to Questions

**Comments to the Author**

1. If the authors have adequately addressed your comments raised in a previous round of review and you feel that this manuscript is now acceptable for publication, you may indicate that here to bypass the “Comments to the Author” section, enter your conflict of interest statement in the “Confidential to Editor” section, and submit your "Accept" recommendation.

Reviewer #1: (No Response)

Reviewer #3: All comments have been addressed

2. Is the manuscript technically sound, and do the data support the conclusions?

Reviewer #1: (No Response)

Reviewer #3: Yes

3. Has the statistical analysis been performed appropriately and rigorously? 

Reviewer #1: (No Response)

Reviewer #3: Yes

4. Have the authors made all data underlying the findings in their manuscript fully available?

Reviewer #1: (No Response)

Reviewer #3: Yes

5. Is the manuscript presented in an intelligible fashion and written in standard English?

Reviewer #1: (No Response)

Reviewer #3: Yes

6. Review Comments to the Author

Reviewer #1: Thank you for the revised manuscript and detailed responses to my previous suggestions. I find the revised manuscript to be much more clear and more comprehensible. Because you addressed my previous suggestions, I find the manuscript ready to be published.

Reviewer #3: (No Response)

7. PLOS authors have the option to publish the peer review history of their article (what does this mean?). If published, this will include your full peer review and any attached files.

Reviewer #1: No

Reviewer #3: **Yes: **Burcu KÜÇÜKKAYA

---

## [Editor Report · Acceptance letter]

19 Dec 2023

PONE-D-23-11358R1 

PLOS ONE

Dear Dr. Durmaz, 

I'm pleased to inform you that your manuscript has been deemed suitable for publication in PLOS ONE. Congratulations! Your manuscript is now being handed over to our production team.

Kind regards, 

on behalf of

Professor Sylvester Chidi Chima 

Academic Editor

PLOS ONE